# Meta-Analysis of Yields of Crops Fertilized with Compost Tea and Anaerobic Digestate

Franco Curadelli [1], Marcelo Alberto [1], Ernesto Martín Uliarte [2], Mariana Combina [2,3] and Iván Funes-Pinter [2,3,*]

1 Facultad de Ciencias Agrarias, Almirante Brown 500, Universidad Nacional de Cuyo, Chacras de Coria, Mendoza M5528AHB, Argentina
2 Instituto Nacional de Tecnología Agropecuaria, Estación Experimental Mendoza (INTA EEA Mendoza), San Martin 3853, Chacras de Coria, Mendoza M5507EVY, Argentina
3 Consejo Nacional de Investigaciones Científicas y Técnicas (CONICET), Av. Ruiz Leal s/n—Parque Gral. San Martín, Mendoza M5500XAD, Argentina
* Correspondence: ifunes.pinter@inta.gob.ar

**Abstract:** Organic inputs constitute an alternative way to replace or reduce the use of agrochemicals in order to increase sustainability and reduce negative impacts of agriculture on the environment. A consistent determination of average yields obtained with organic fertilizers in comparison to synthetic fertilizers is necessary to assess their potential in both commercial and organic agriculture. To achieve this goal, a meta-analysis of existing scientific data of yields obtained with digestate or compost tea fertilization was performed. After a systematic bibliographic search of scientific publications, 35 final papers remained from >1000 initial results. Data of crop yield with digestate or compost tea fertilization, as well as control and synthetic fertilizer treatments, were extracted from the selected articles and used to calculate response ratios (ratios of means), obtaining 106 observations. The meta-analysis showed that digestate fertilization produced yields 80% higher than the control. Yields were statistically similar to those obtained with conventional treatments with chemical fertilization (only 2% lower in average). The results for digestate are considered robust as the significance did not change after conducting publication bias analyses. However, the high heterogeneity observed suggests the existence of explanatory variables accounting for part of the observed dispersion. Subgroup analyses were conducted to determine the variation of the results of digestate across crop species and field or greenhouse experiments, while compost teas did not present adequate data quantity to obtain reliable results. According to this meta-analysis, anaerobic digestate had an acceptable performance as fertilizer for several crops at a reported application rate ranging from 100 to 480 kg N ha$^{-1}$. Regarding compost tea, yields were 92% higher than the control and 10% lower than synthetic fertilization treatments, although more information is required to draw a conclusive result due to the low number of observations of this fertilizer. Further investigation is necessary to understand yield variations under different scenarios to study the influence of secondary variables and to propose management measures for producers.

**Keywords:** bio-input; crop yield; organic fertilizer; response ratio

## 1. Introduction

In recent decades, the use of synthetic fertilizers has been crucial for high-production agricultural systems [1]. However, despite the high yields reached in the short term, their indiscriminate use can trigger several negative environmental effects, such as loss of soil organic matter, nutrient lixiviation, greenhouse gas emissions and deterioration of soil quality [2]. World grain production grew four times between 1960 and 2015 with only a minor increase in the surface of land destined to these crops, while the quantity of agrochemicals used multiplied by nine. This fact suggests that a certain increment in agricultural production requires an even higher increase in the amount of agrochemicals consumed [3].

Around the world, efforts are being made to implement organic products as alternatives to synthetic inputs in agriculture in order to increase sustainability and reduce negative impacts on the environment and human health [4]. Bio-inputs are products based on active organisms or their metabolites that have properties of plant growth promotion, control of plant diseases or improvement of soil conditions. Their production costs tend to be lower and, in many cases, their elaboration implies using, reducing and treating organic residues [5–8].

Among all bio-inputs, compost teas have become relevant due to their plague control and plant growth promotion properties, associated with their beneficial microorganisms and nutrient content. The general production method consists in brewing compost in free-chloride water for a period of 2 to 10 days, depending on whether the process is aerated or non-aerated, respectively. The brewed product is then filtered to obtain the liquid fraction [9–12].

Another promising organic fertilizer is liquid digestate, also known as bioslurry, which is produced by the anaerobic digestion of organic material, mainly as a by-product of biogas generation [13,14]. Both types of bio-products are generated from organic residues, either fresh (digestate) or previously composted (compost tea), and have high potential as plant fertilizers. However, scientific literature regarding compost teas focuses mostly on their biocontrol properties, while in the case of anaerobic digestate, its fertilization properties are highlighted [15–17].

In this scenario, it becomes necessary to study the yields of crops fertilized with these bio-inputs and to compare them with those obtained fertilizing with conventional synthetic products, in order to provide solid knowledge for developing more sustainable alternatives for agricultural production [10]. Although a narrative review with vote counting is a traditional tool in these cases, its conclusions can be imprecise or even erroneous when aiming to determine the existence and magnitude of an average effect [18]. Thus, it is not a proper approach to analyze the effect of organic fertilizers in yields.

To our knowledge, studies conducted so far that assessed the effects and properties of digestate or compost tea fertilization consist in scientific or technic literature reviews. In the case of digestate, previous works state that digestate fertilization has a generally positive effect compared to no fertilization [14]. The existing literature agrees that digestate can produce similar yields than commercial fertilizers [19,20], depending on crop type and the level of synthetic fertilizer replacement. However, these statements derive from individual cases or experiments with methodological shortcomings [21]. Another review cites a case where plants fertilized with compost tea showed a nutritional status similar to conventional fertilizer treatments [22].

Meta-analysis is a powerful tool that makes it possible to summarize evidence from multiple scientific reports by the application of quantitative statistical methods, collecting and combining qualitative and quantitative relevant data to obtain statistically robust conclusions. Besides meta-analysis being considered an evidence-based resource, not all publications provide adequate data for the analysis, since some of them show heterogeneity in criteria, population size, etc. [23]. Related to this, the most usual handicaps are the difficult and time-consuming identification of appropriate studies, the possible occurrence of publication bias and the need of advanced statistical techniques. Despite the drawbacks, meta-analysis allows the researcher to calculate the average magnitude of an effect as well as the heterogeneity in the data with their corresponding confidence levels, inferring general conclusions [24,25].

Although several reviews and meta-analyses have compared crop yields obtained with organic fertilizers against conventional chemical fertilization [1,2,26], results of multiple organic fertilizers were often grouped together and analyzed in a nonspecific way. These papers combined the results of animal manure, green manure, digestates, and other organic fertilizers in a few general categories, although different types of bio-inputs can produce different effects on plant productivity, as they differ in their properties, such as C/N ratio, N, P and organic matter content, electrical conductivity, and pH [27]. In several cases,

organic fertilization categories in these reports included treatments combining organic and synthetic fertilization, hindering the possibility of identifying the specific effect of organic products.

For all of these reasons, the aim of the present work was to perform a meta-analysis of crop yields fertilized with compost tea or anaerobic digestate, based on available scientific reports published in the last decade. It is noticeable that the aforementioned antecedents did not provide a quantitative measure of the general effect of digestate or compost tea, and based their conclusions on a low number of observations. The novelty of our work is that we used meta-analysis as a tool to obtain a quantitative result, determining the general effect of these organic fertilizers on crop yields from a large number of published scientific observations. Although the exclusion of some technic literature reduced the number of eligible experiments, it improved the reliability on the results.

Considering that there are no meta-analyses reporting the use of these two bioproducts on crop production, we aimed to isolate their individual effects as bio-fertilizers in order to prove the potential of these products. Thus, we compared crop yields obtained with these bio-inputs with respect to conventional fertilization and water control treatments through the analysis of yield response ratios. Additionally, we conducted a subgroup-analysis to determine variations in the effect across different crop species, or between experiments conducted in field or greenhouse.

## 2. Methodology

### 2.1. Systematic Bibliographic Search

A systematic bibliographic search was performed to obtain peer-reviewed articles published in English or in Spanish between the years 2010 and 2020. The terms "compost tea", "bioproduct", "biogas slurry", "biosludge", "bioslurry", "compost extract", "culture", "digestate", "fertilizer" and "organic fertilizer" were searched in the Title, Abstract and Keyword fields, either alone or combined by logical operators in order to produce more specific results.

The scientific databases comprised in the search were SciELO (https://scielo.org/es/ (accessed on 26 July 2020)), Science Direct (https://www.sciencedirect.com/ (accessed on 5 September 2020)) and NCBI—PMC (https://www.ncbi.nlm.nih.gov/pmc/advanced (accessed on 10 July 2020)). A secondary search was performed in Research Gate (https://www.researchgate.net/ (accessed on 8 November 2020)) and Google Scholar (https://scholar.google.es/ (accessed on 8 November 2020)) databases to find potentially useful articles mentioned in the papers obtained by the primary search.

The bibliographic search produced approximately 1200 scientific papers, which were reduced to a preselection of 194 articles after screening titles and abstracts. At a third stage, preselected papers were read in full length, obtaining a final dataset of 35 articles which complied with the selection criteria.

### 2.2. Selection Criteria

Only peer-reviewed articles reporting primary data were included in the meta-analysis. When two or more papers were based on the same experiment, only the most relevant or most complete was selected.

Selection criteria consisted in the following:

- Papers based on scientific experiments including at least one treatment of pure fertilization with anaerobic digestate or compost tea (not combined with any other organic or synthetic fertilizer);
- Papers based on experiments including a control treatment (no fertilizer), and optionally a pure synthetic fertilizer treatment (not combined with organic fertilizers).
- Papers based on field, greenhouse, and pot experiments were selected (papers based on in vitro experiments were discarded);
- Papers based on experiments lasting at least 3 weeks or longer;

- Papers informing average crop yield value for each treatment, along with individual or pooled dispersion measures;
- Papers that measured crop yield as biomass: dry or fresh weight of either the total plant, aerial part or the organ of agronomic interest (e.g.,: fruit). When both dry and fresh weights were reported, the former was selected. Other measures, such as leaf area or number of fruits, were discarded.

### 2.3. Meta-Analysis

A meta-analysis of yields of crops fertilized with anaerobic digestate or compost tea was performed in order to assess the effects of these bio-inputs in comparison to the control treatment and synthetic fertilizer treatments. Meta-analysis consists in the application of quantitative statistics to summarize evidence from a set of studies. This methodology includes tools for the calculation of central tendency and heterogeneity measures for a group of observations obtained from different studies, allowing inference of the magnitude and significance of the real effect of a treatment [28].

Subgroup analyses were conducted for digestate/control and digestate/synthetic fertilizer observations, applying a random-effects model as described in Harrer et al. (2019a) [29]. The pooled subgroup RR values were considered significantly different from each other if $p < 0.05$ following the chi-square test. The grouping variables were trial environment (field or greenhouse) and crop species. Categories with less than 3 observations were excluded from the subgroup analyses. Compost tea data was not included as the low number of observations made impossible the categorization.

The outcome measure analyzed in a meta-analysis is usually called "effect size" (abbreviated "ES" [30]. The effect size selected for this meta-analysis was the ratio of means or response ratio (RR), calculated according to Equation (2) [25]:

$$RR = \bar{y}_t / \bar{y}_c \tag{1}$$

where "$\bar{y}_t$" represents the average yield of the treatment group (organic fertilizer), and "$\bar{y}_c$" represents the average yield of the corresponding reference group (control or conventional fertilizer).

This index is applicable to outcomes measured on a true ratio scale (such as mass), with positive values and an absolute zero. The main advantages of RR are its simplicity of calculus, its physical/biological meaning which quantifies the proportional change resulting from an experimental manipulation, and its non-dimensional and standardized value [18].

Two separate meta-analyses were performed for the following datasets:

- Dataset 1: RR of average yield obtained by fertilization with digestate or compost tea, in relation with the average yield of the corresponding control treatment (same experimental conditions except fertilization);
- Dataset 2: RR of average yield obtained by fertilization with digestate or compost tea, in relation with the average yield produced by application of conventional fertilizer (same experimental conditions except fertilization).

### 2.3.1. Data Extraction

The selected papers displayed their experimental results either in text, in tables or graphically. In the latter case, values were digitized using the web tool WebPlotDigitizer (https://automeris.io/WebPlotDigitizer/ (accessed on 23 April 2021) [31]).

Within a scientific paper, an experiment was defined as each experience conducted at a single site, consisting of a single crop grown in homogeneous conditions (field or greenhouse, soil type, irrigation, duration, planting and harvest dates, etc.), where treatments varied only in fertilizer type and rate. As most articles described the results of two or more experiments, it was necessary to determine the unit of observation. It was defined as each single pair of treatment (digestate or compost tea) and reference groups (control

or conventional fertilization) within the same experiment, from which one value of yield response ratio was calculated (see example in Figure 1). Besides average yields, sample sizes and measures of dispersion (variance, standard deviation or standard error) of both treatment and reference groups were also extracted.

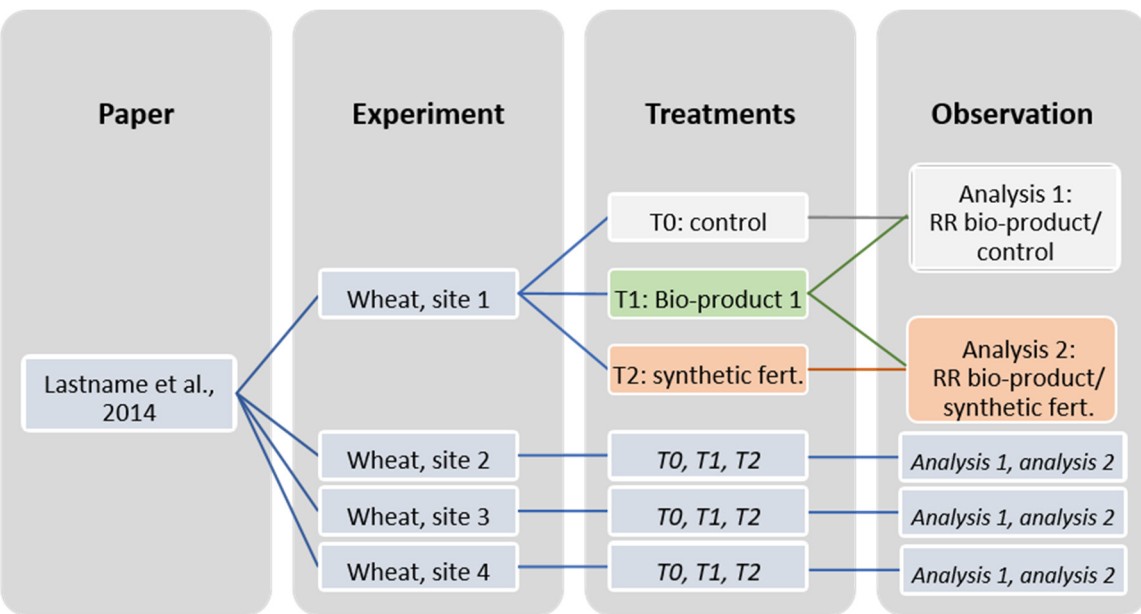

**Figure 1.** Obtention of response ratio (RR) observations from selected articles to evaluate bio-products performance.

Some articles reported only the results of analyses of variance (ANOVA), grouping treatment means by their statistically similar values but informing no measure of dispersion for individual treatments. Other meta-analyses have solved this issue by estimating dispersion from similar experiments within the dataset [26], using the average variance of the whole dataset [27], or calculating the pooled variance for all the treatments in the experiment [2]. The latter approach was applied in the present investigation, calculating the pooled variance from the ANOVA results, considering that homoscedasticity and normality assumptions of the data were fulfilled. This practice allows for the inclusion of a larger number of observations, at the expense of losing precision.

The study of the influence of nitrogen application rate, crop type, or source materials of the compost tea or anaerobic digestate was beyond the scope of the present investigation. When two or more treatments consisted in the same fertilizer type, varying only the application rate, the treatment with the highest yield was included in the analysis and the others were discarded since these differences would be attributable to suboptimal application rates rather than to a property of the fertilizer itself.

Whenever a single experiment presented two or more similar treatments included in the scope of this investigation (e.g., two treatments fertilized with anaerobic digestate from different sources), their values were pooled. This was necessary to prevent the unit-of-analysis error as two or more effect sizes calculated with the same reference value would be codependent, violating the assumption of independence of observations. In these cases, average yields, sample sizes and standard deviations were combined prior to the calculus of the RR observation, in accordance with Harrer et al. (2019) [29].

A preliminary number of 106 yield values of organic fertilization treatments was reduced to 69 by pooling results of similar treatments within each experiment. The final dataset for the analysis of organic fertilizer/control RR consisted of 69 observations from 69 experiments described in 35 articles. For the analysis of organic/synthetic fertilizer RR, the dataset comprised 52 observations from 52 experiments reported in

26 studies. Observations of anaerobic digestate and compost tea fertilization treatments were handled separately.

### 2.3.2. Basic Formulae

Response ratios were transformed to the natural logarithm of RR (hereon represented as "L") to conduct the analyses. The transformation was applied for two reasons: first, to linearize the metric, as the new variable is equally affected by deviations in the numerator and the denominator of the original RR; second, the sampling distribution of RR is skewed, while the distribution of its logarithm is much more normal in small samples.

The formulae for the calculus of each observation of L (Equation (2)) and its variance (Equation (3)) were extracted from Hedges et al. (1999) [25]:

o    Equation (2): observation of L

$$L = \ln(RR) = \ln(\bar{y}_t) - \ln(\bar{y}_c) \tag{2}$$

- Equation (3): Variance of L:

$$V_L = (SD_t)^2 / (n_t \, \bar{y}_t^2) + (SD_c)^2 / (n_c \, \bar{y}_c^2) \tag{3}$$

where "$\bar{y}$" is the average yield resulting from an experiment, "n" is the sample size and "SD" the standard deviation of the group. Subscripts "t" and "c" denote treatment and reference groups, respectively.

### 2.3.3. Preliminary Analyses

GOSH plot analysis [32] and influence analysis were conducted to identify possible outliers or subgroups of observations, as described in Harrer et al. (2019) [29]. No subgroups were detected, according to the GOSH plots. Only two observations of the digestate/control dataset were identified as outliers due to their high values of RR. Thus, the final meta-analysis for this comparison excluded these observations, reducing the number of observations from 61 to 59, in order to produce a more conservative and precise estimate of the pooled effect size. No outliers were detected in the other datasets.

### 2.3.4. Software

Data tabulation and calculation were performed in Excel (Microsoft Office). Numeric transformation from graphical values was performed by the web tool WebPlotDigitizer [31]. All the preliminary, meta-analysis and publication bias analyses were conducted in R (R Core Team, 2020 [33], Figure 2), using the meta [34], metafor [35] and dmetar [36] packages.

Other R packages used for data manipulation, graphics and export of the results were: dplyr [37], readxl [38], ggplot2 [39], rmarkdown [40], bookdown [41], kableExtra [42], and knitr [43].

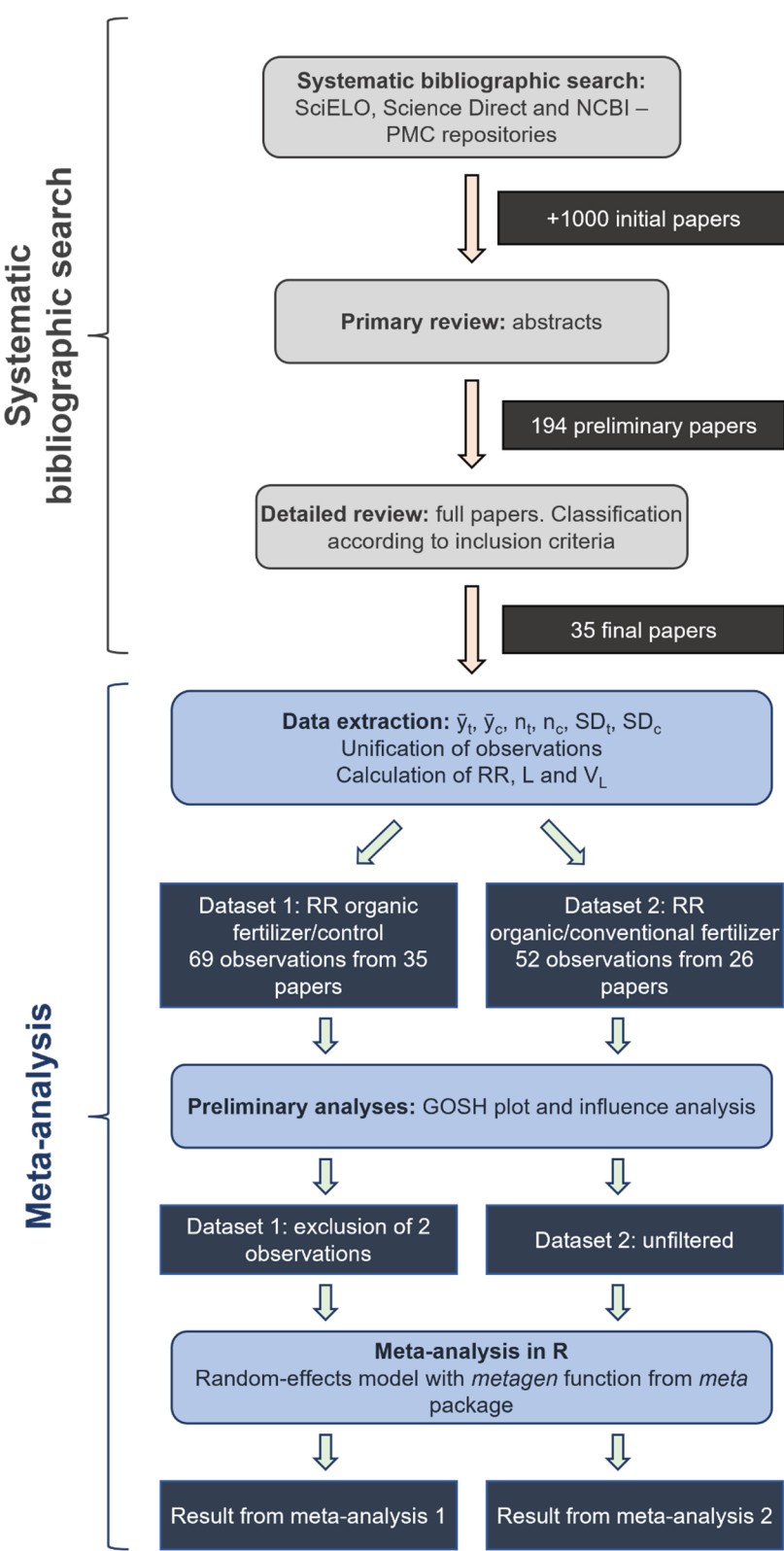

**Figure 2.** Graphical summary of the meta-analysis investigation process to evaluate bio-products performance in crop production.

## 3. Theory/Calculation

### 3.1. Meta-Analytic Model

The overall effect size was estimated from L observations using a Random Effects Model. This type of meta-analytic model considers that individual effect sizes represent a set of different populations with their own true effect sizes, and the model aims to estimate the overall mean of true effect sizes across populations [36]. This model also provides an estimate of between-population variance, $\tau^2$.

The model assigns inverse-variance weights to each observation to produce a more precise estimate of the overall effect size. Observations with high dispersion (low precision) will be assigned a lower weight than those with low dispersion [18].

Formulae for the calculation of the overall effect size are described in Hedges et al. (1999) [25], Equation (4).

$$L^* = \frac{\sum_{j=1}^{k} W_j \, L_j}{\sum_{j=1}^{k} W_j} \tag{4}$$

Inverse-variance observation weight (Equation (5)):

$$W_j = \frac{1}{V_{Lj} + T^2} \tag{5}$$

where "$L_j$" is the j-th observation of log response ratio, "k" is the total number of observations, "$L^*$" is the weighted average of L, "$W_j$" is each observation weight, "$V_{Lj}$" is the observed variance of $L_j$, and "$T^2$" is the estimator of $\tau^2$, the between-study variance. The inverse-variance weight of each observation includes both the within-study (observed) variance and the between-study variance.

The resulting overall value of log RR and its corresponding confidence interval limits were transformed back to the original metric for their interpretation as response ratios.

### 3.2. Estimation of Between-Study Variance

There are several methods for calculating $T^2$, the estimator of between-study variance ($\tau^2$). A common method is the DerSimonian–Laird estimator, given its relative simplicity of calculus and interpretation. However, it tends to underestimate heterogeneity, producing narrower confidence intervals. Some alternative methods generate more accurate results, such as the Hartung–Knapp–Sidik–Jonkman method (HKSJ, [31]), the Restricted Maximum Likelihood method (REML, [18]), and the Mandel–Paule method (MP, [44]).

According to simulations conducted by Bakbergenuly et al. (2020) [45], the accuracy of estimation of $\tau^2$ in meta-analysis of L might be limited depending on the sample size, underlying data distribution and real values of L and $\tau^2$. In the referred study, MP showed reasonably good results, but REML also behaved as a good estimator in some of the models. Additionally, the latter is favored by some statisticians, so REML was the selected method for the present meta-analysis.

Besides $T^2$, Q and $I^2$ statistics were used for assessing between-study heterogeneity in the results. The equations are displayed below:

- Cochran's Q: it is a standardized measure calculated as the sum of the squares of the difference between each observed effect size ($ES_j$) and the fixed-effect model estimate of the overall effect size (M), weighted by the inverse of each study's variance ($SD_j^2$). This value was used in the influence analyses to determine which observations had greater contribution to overall heterogeneity. Its formula, according to Borenstein (2009) [18], can be written as Equation (6).

$$Q = \sum_{j=1}^{k} \left( \frac{ES_j - M}{SD_j} \right)^2 \tag{6}$$

- Higgin's and Thompson's $I^2$: represents the percentage of variability in the effect sizes which is not caused by sampling error [29]. It is derived from Q, with the following formula (Equation (7)):

$$I^2 = \frac{Q - df}{Q} * 100\%$$
$$df = k - 1$$

(7)

where "df" is the number of degrees of freedom and "k" is the number of studies. Being a percentage, $I^2$ is useful to contrast heterogeneity across analyses.

### 3.3. Subgroup Analyses

After the general meta-analysis, digestate observations were partitioned by crop species and by environment of the experiments to determine if effects varied between the categories of each variable. Subgroup analyses were conducted for digestate/control and digestate/synthetic fertilizer observations, applying a Random Effects Model as described in Harrer et al. (2019) [29]. Compost tea data was not included in the analysis, as the low number of observations made impossible the partition of the dataset.

Every subgroup analysis consists of two parts: (1) pooling the effect of each subgroup, and (2) comparing the effects of the subgroup. Random-effects models were applied at both levels. The pooled subgroup RR values were considered significantly different from each other if $p < 0.05$ following a chi-square test. Categories with k < 3 observations were excluded from the subgroup analyses. Heterogeneity data for groups with k < 5 observations is considered not reliable [18], and thus were not displayed in the resulting forest plots.

### 3.4. Publication Bias

Publication bias analyses were performed to determine the existence of publication bias in the datasets. These included Egger's test [46] and the Duval's and Tweedie's Trim and Fill procedure [47]. Egger's test quantifies the significance of asymmetry in the observations attributable to the existence of publication bias. Duval's and Tweedie's Trim and Fill procedure calculates a conservative, "unbiased" estimate of the pooled effect size and its confidence interval by adding modeled observations in the side of the distribution where unpublished results would be expected to be. In this case, modeled observations corresponded to low effect sizes, assuming that these results would be less likely to be published.

## 4. Results

### 4.1. Analysis 1: Organic Fertilizer/Control Response Ratio

#### 4.1.1. Dataset 1 Observations

The systematic bibliographic search produced a total of 69 observations of organic fertilizer/control RR that satisfied the selection criteria. Included experiments comprise 25 different crop species, being wheat the most represented (19% of the observations), followed by corn (14%) and tomato (9%). Five other crops had 3 to 5 observations each, representing 28% of the dataset, and the remaining 17 crops presented less than 3 observations, accounting for 30% of the data (Figure 3).

Figure 4a shows that 52% of dataset 1 comes from digestate field assays and 36% from digestate greenhouse assays, totalizing 88% of digestate observations. The remaining 12% corresponds to compost tea greenhouse observations, as no compost tea field observations satisfied the selection criteria. Approximately 70% of the dataset comes from experiments that lasted more than 2 months, completing the crop cycle in several cases (Figure 4b). The remaining 30% is evenly distributed between observations from experiments lasting less than one month, and those lasting up to two months.

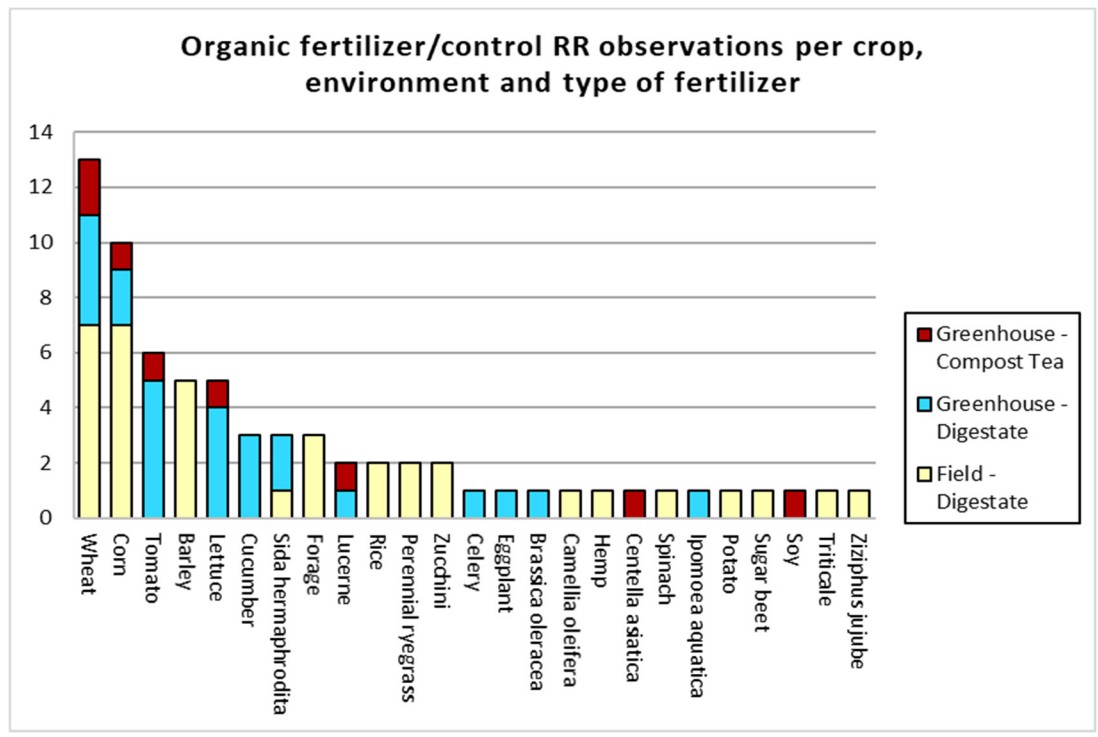

**Figure 3.** Distribution of dataset 1 observations classified by crop species used in meta-analysis evaluating bio-products performance.

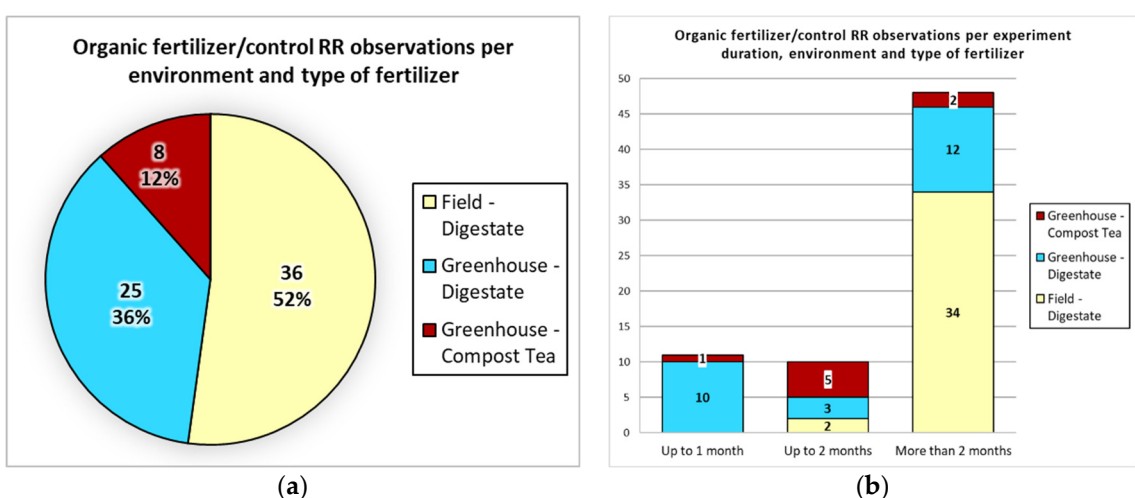

| (a) | (b) |

**Figure 4.** Distribution of dataset 1 observations according to type of organic fertilizer and environment (**a**). Distribution of dataset 1 observations according to type of organic fertilizer and duration of the experiment (**b**).

### 4.1.2. Meta-Analysis of RR

The meta-analysis of digestate/control RR resulted in a pooled ES of 1.80, based on k = 59 observations. The lower and upper limits of the 95% confidence interval were 1.58 and 2.04, respectively, representing a significant difference between digestate and control treatments. This result is conservative, as it was obtained after removing 2 atypical observations with highly positive influence on the pooled value, which were highlighted by the influence analysis. The pooled ES for compost tea/control RR obtained a value of 1.92, with k = 8 observations. Its 95% confidence interval ranged from 1.49 to 2.47 (Figure 5).

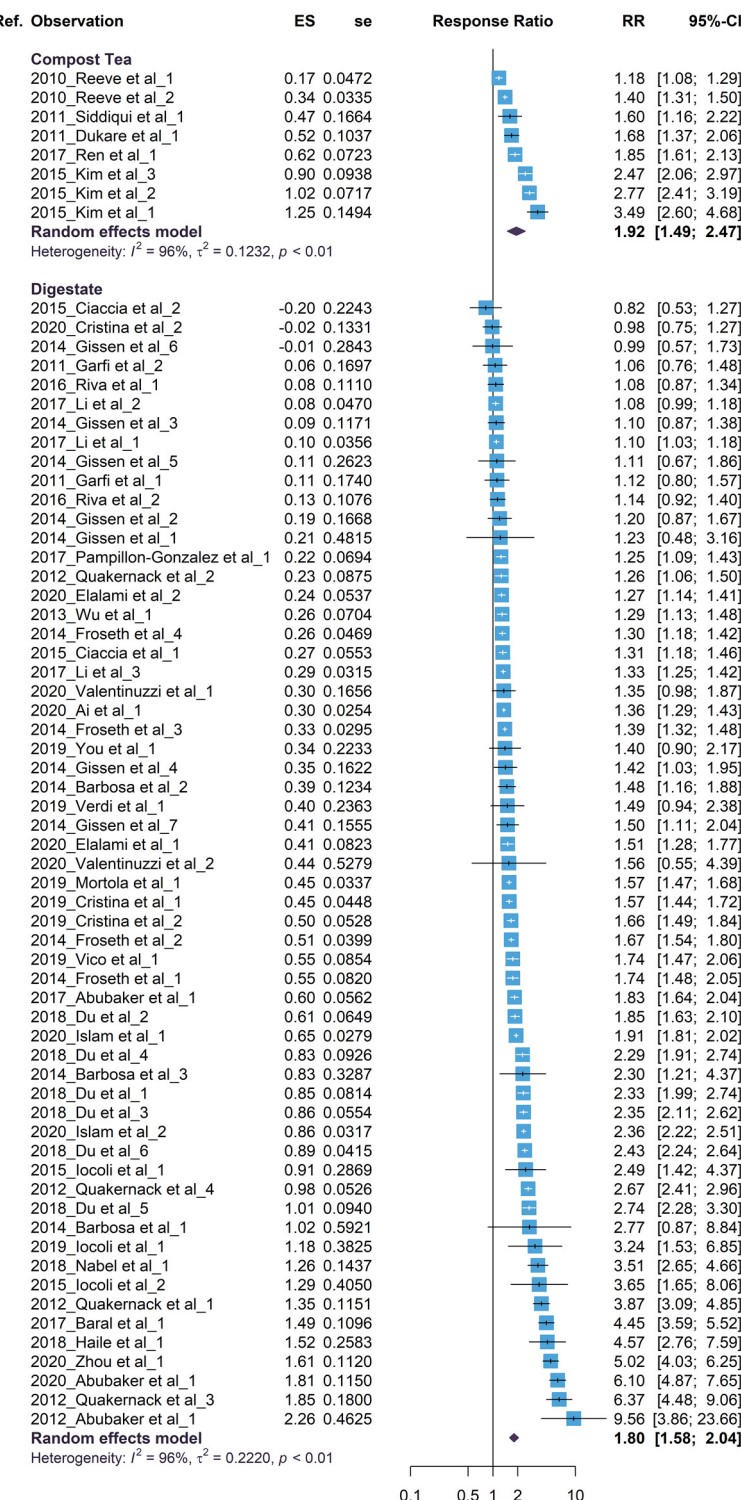

**Figure 5.** Forest plot of the meta-analysis of organic fertilizer/control response ratio (RR). Numeric RR values and 95% confidence intervals (95%-CI) are expressed in the original metric (ratio). Analyzed effect sizes (ES), standard errors (se) and the graphic scale (horizontal axis) are logarithmic to represent the transformation of the data [7,11–13,15,16,48–76].

$I^2$ values were high in both cases: 95.8% for compost tea and 96.1% for digestate. As the meta-analysis considered a Random-Effects Model, each pooled ES represents an estimation of the average ES for a range of statistical populations (different crops,

environments, organic fertilizer qualities, etc.). It does not estimate the mean value of any specific population, with definite experimental conditions.

### 4.1.3. Subgroup Analyses of Digestate/Control Yield RR

The effect on yield of digestate fertilization compared to control treatments showed significant differences between crop species, being wheat and lettuce the groups with higher RR values. Most cultures presented 5 observations or less, except for wheat and corn (Figure 6).

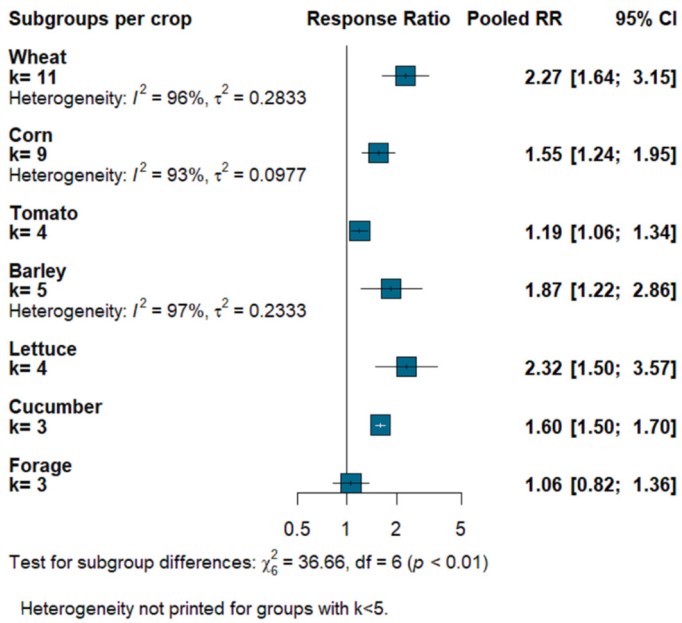

**Figure 6.** Effect of digestate fertilization on yield respect to control treatments for different crop species.

No differences were detected between field and greenhouse experiments, although the pooled value was moderately higher in the second group (Figure 7). In both cases, yields obtained with digestate fertilization were significantly higher than the control treatments. It is important to mention that the number of observations was higher than 5 in both cases.

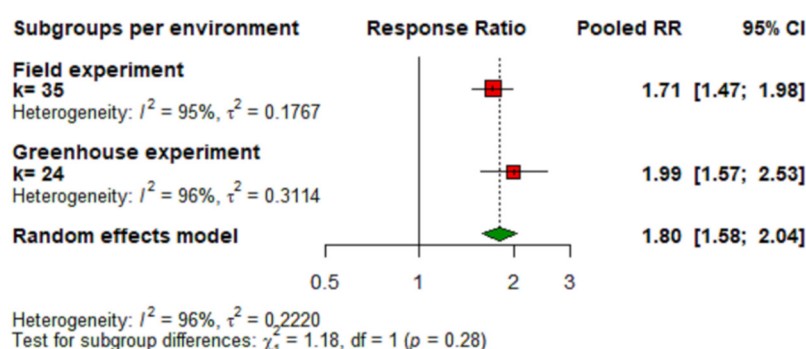

**Figure 7.** Effect of digestate fertilization on yield respect to control treatments for experiments conducted in different environments.

### 4.2. Analysis 2: Organic Fertilizer/Synthetic Fertilizer Response Ratio

#### 4.2.1. Dataset 2 Observations

The systematic bibliographic search produced a total of 52 observations of organic/synthetic fertilizer RR that matched the selection criteria. Included experiments comprise 20 dif-

ferent crop species, the most represented of which were wheat (23% of the observations), corn (15%) and barley (10%). The other 3 crops had 3 observations, comprising 17% of the data, while the remaining 14 crops presented less than 3 observations (35% of the dataset; Figure 8).

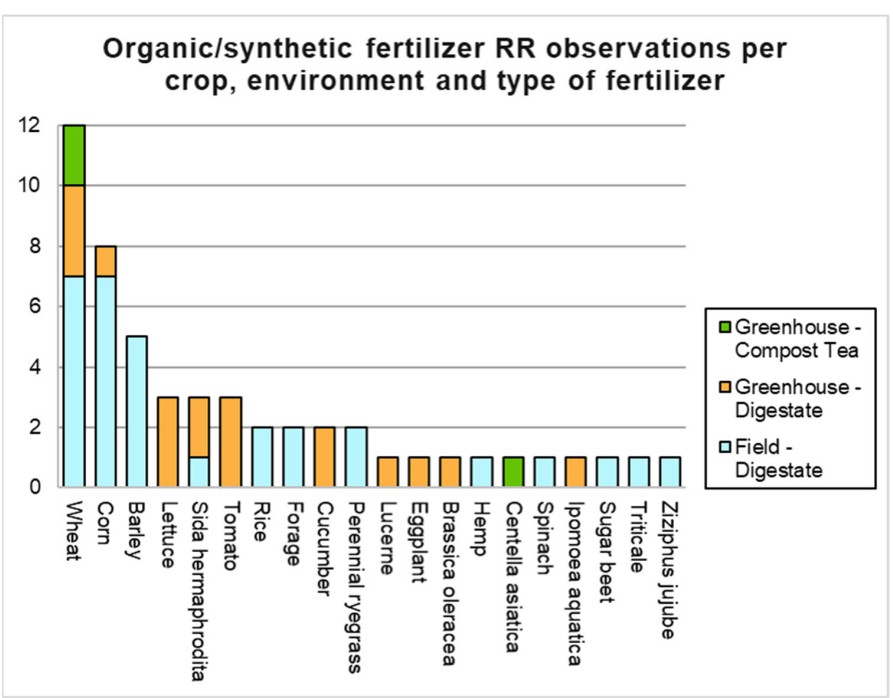

**Figure 8.** Distribution of dataset 2 observations classified by crop species grown by compost tea and bioslurry application.

As shown in Figure 9a, 94% of dataset 2 observations correspond to digestate. Digestate field assays represent 59% of the dataset, and an additional 35% comes from digestate greenhouse assays. All compost tea observations proceed from greenhouse assays, as none compost tea field experiments satisfied the selection criteria. Regarding experiment duration, 75% of the dataset comes from experiments that lasted more than 2 months (Figure 9b). The remaining observations were obtained by experiments that lasted less than one month (15% of the dataset), or up to two months (10%).

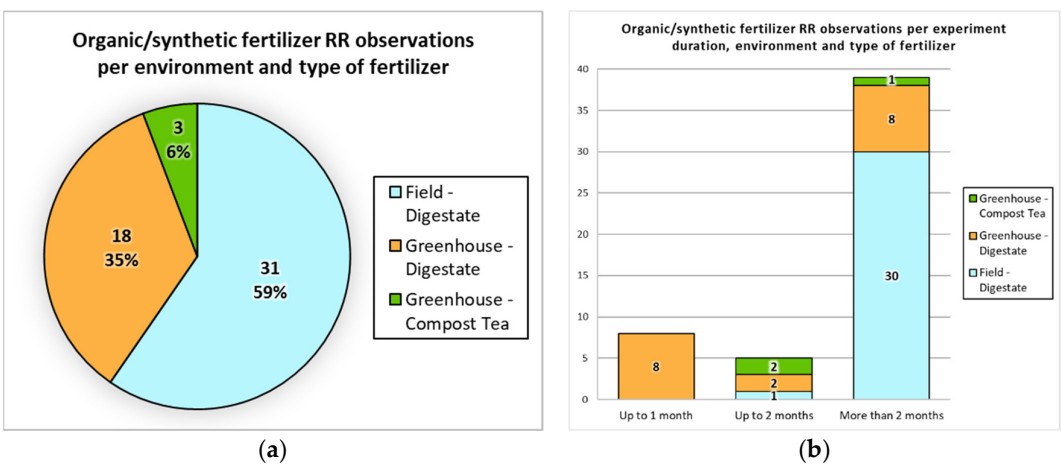

(**a**)　　　　　　　　　　　　　　　(**b**)

**Figure 9.** Distribution of dataset 2 observations according to the type of organic fertilizer and environment (**a**). Distribution of dataset 2 observations according to the type of organic fertilizer and duration of the experiment (**b**).

### 4.2.2. Meta-Analysis of RR

According to the meta-analysis results (Figure 10), the pooled RR for the digestate/synthetic fertilizer comparison was 0.98, based on k = 52 observations. The 95% probability confidence interval ranged from 0.90 to 1.06, representing a non-significant difference between yields obtained with both treatments. Regarding compost tea/synthetic fertilizer RR, the meta-analysis was based on only k = 3 valid observations, producing a pooled ES of 0.90 with a 95% probability confidence interval ranging between 0.70 and 1.17. The non-significant result for compost tea is considered merely informative due to the low size of the dataset from which it was obtained. $I^2$ values were also high for both comparisons, as in the previous cases: 98.4% for compost tea and 95.9% for digestate.

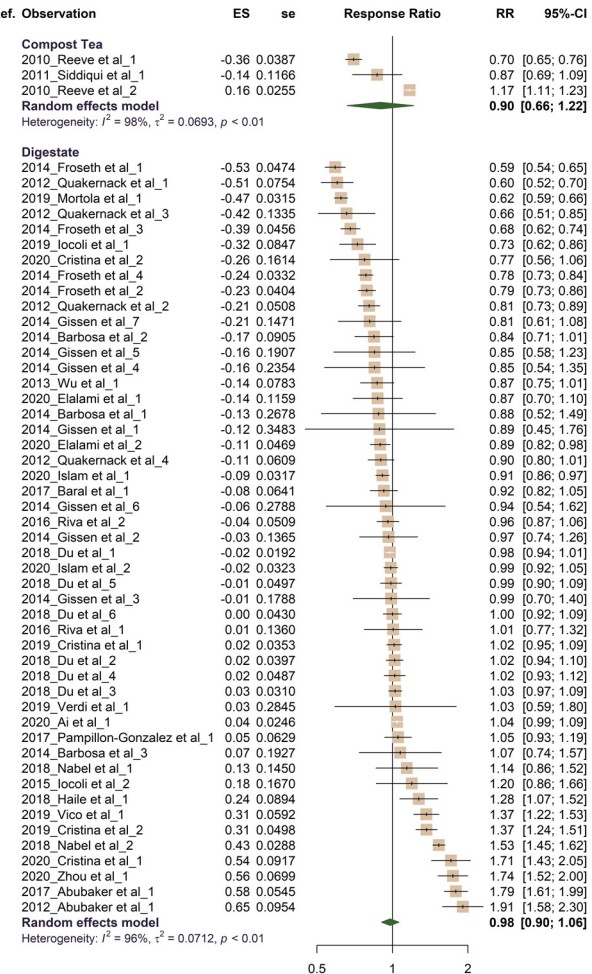

**Figure 10.** Forest plot of the meta-analysis of the response ratio (RR) of organic/synthetic fertilizer. Numeric RR values and 95% confidence intervals (95%-CI) are expressed in the original metric (ratio). Analyzed effect sizes (ES), standard errors (se) and the graphic scale (horizontal axis) are logarithmic to represent the transformation of the data [7,12,13,15,48,51,52,54,56–59,61,63–74,76].

### 4.2.3. Subgroup Analyses of Digestate/Control Yield RR

The effect on yields of digestate fertilization compared to synthetic fertilization varied significantly across crop species. The wheat, corn and barley were the only cultures with 5 or more observations, while the rest of the groups presented k = 3. Yields of wheat and corn were statistically similar between digestate and synthetic fertilizers. In the case of barley, yields obtained with digestate were significantly lower than synthetic fertilization treatments (Figure 11).

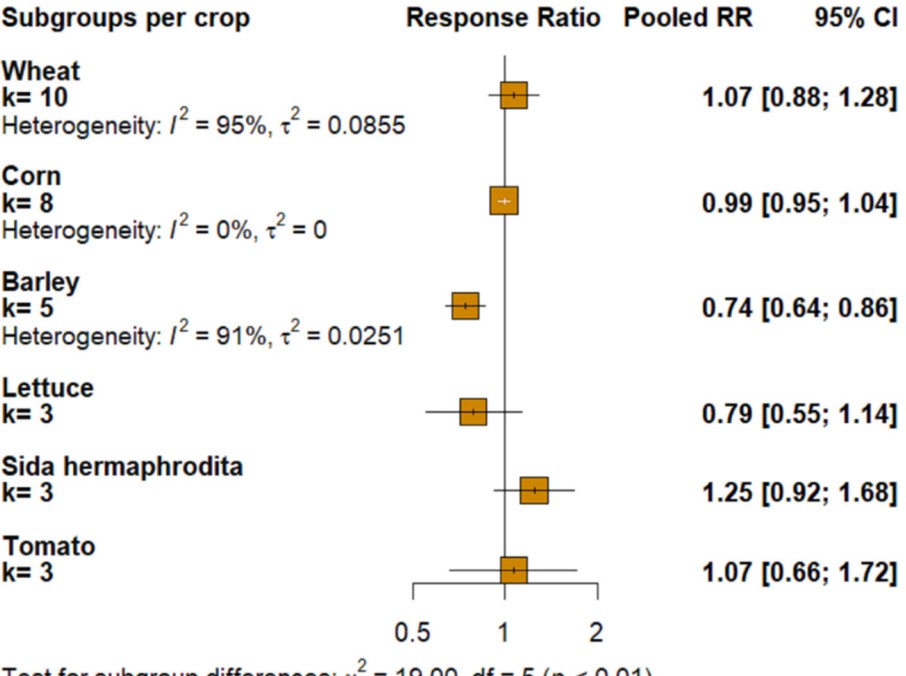

**Figure 11.** Effect of digestate fertilization on yield respect to synthetic fertilization for different crop species.

Yields were statistically similar between digestate and synthetic fertilization treatments in both field and greenhouse experiments, although the result was moderately higher in the second group (Figure 12). This result is robust, as the two groups presented more than 5 observations.

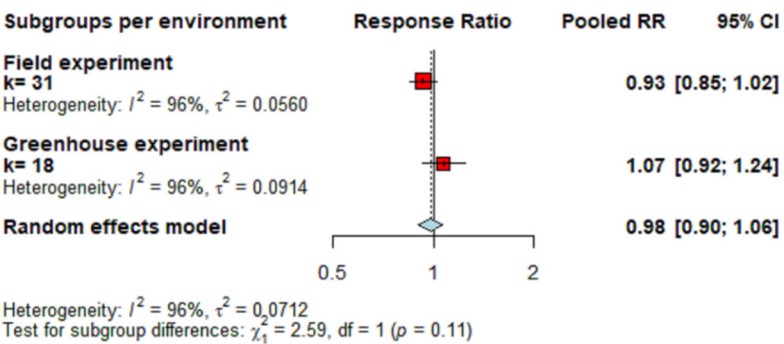

**Figure 12.** Effect of digestate fertilization on yield respect to synthetic fertilization treatments for experiments conducted in different environments.

### 4.3. Publication Bias Analyses

Egger's test and Duval and Tweedie's Trim and Fill procedure were performed to determine the existence of publication bias in the datasets. In all cases, the results of these tests were either non-significant, indicating that there is no concrete evidence of the existence of publication bias, or not conclusive, due to an insufficient number of observations.

#### 4.3.1. Egger's Test

All Egger's tests (Table 1) produced non-significant results, meaning that there is no substantial asymmetry. This result points to the absence of publication bias in the digestate

datasets. In the case of compost tea, the test is based on a low number of observations, meaning it is not reliable due to poor statistical power.

**Table 1.** Results of Egger's test for the different comparisons.

| Response Ratio | *p*-Value | Result | Interpretation |
| --- | --- | --- | --- |
| Digestate/control | 0.161 | Non-significant | The test does not indicate the existence of substantial asymmetry. |
| Compost tea/control | 0.079 | Non-significant | Insufficient data. |
| Digestate/synthetic fertilizer | 0.922 | Non-significant | The test does not indicate the existence of substantial asymmetry. |
| Compost tea/synthetic fertilizer | 0.697 | Non-significant | Insufficient data. |

### 4.3.2. Trim and Fill

The Trim and Fill procedures generated conservative, "unbiased" estimates for the pooled effects of digestate or compost tea yields against control or synthetic fertilizer treatments. In the case of digestate (Figure 13a,b), the value of the pooled estimate of digestate/control RR reduced to 1.53 but the significance of the result remained. Similarly, the pooled digestate/synthetic fertilizer RR decreased to 0.93 in the trim and fill estimate, although the difference between treatments was still not significant. Regarding compost tea (Figure 13c,d), RR for both comparisons remained unchanged as no modeled observations were added, although this is attributable to the low number of observations and subsequent low power of the analyses.

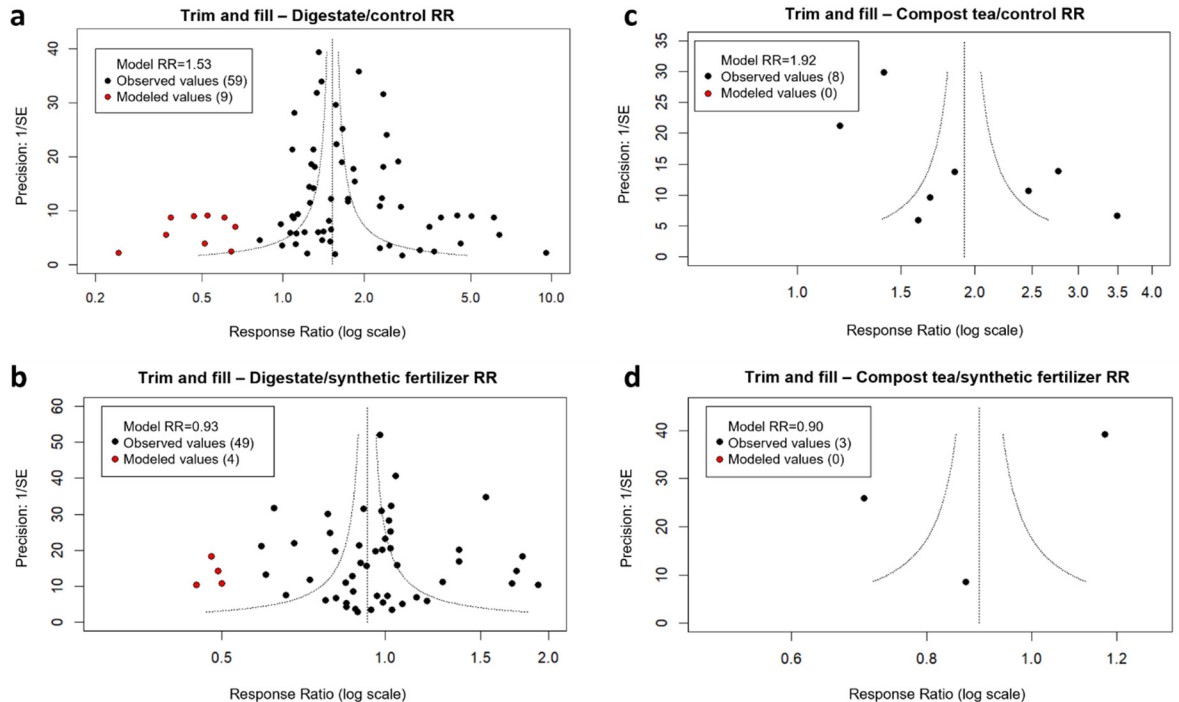

**Figure 13.** Trim and fill funnel plots for organic fertilizer/control (**a**,**c**) and organic/synthetic fertilizer RR observations (**b**,**d**).

## 5. Discussion

In the present study, a meta-analysis of the effects of fertilization with compost tea and anaerobic digestate on crop biomass yield was performed to assess their potential in agricultural production. Despite the high number of initial results yielded by the bibliographic search (1195 scientific papers), most of them were excluded for not fulfilling the predefined selection criteria. For this reason, the final number of meta-analyzed papers

was 35, 30 of which corresponded to digestate and 5 to compost tea. In the second case, the low number of papers suitable for the meta-analysis was mainly explained by the lack of studies focused on fertilization with pure compost tea: most of them studied only its effect as an organic pesticide, or analyzed the nutritional effect in combination with commercial fertilizers.

According to this meta-analysis, digestate fertilization increased biomass yield by 80% on average in respect to control treatments, while compost tea showed an increase of 92% in yield compared to control treatments. Both values are higher than the 70% yield increase reported by Wang et al. (2020) [2] for winter wheat yield with organic fertilizers. Increments in crop yield were foreseeable when comparing these organic fertilizers against control, as control treatments had no fertilization at all. Additionally, the fact that none of the RR observations were significantly negative indicates that the bio-products did not cause significant negative effects such as phytotoxicity, which is regarded as a possible effect by some scientific sources [20,77]. However, the result for compost tea is considered merely informative, due to the low number of observations from which it was obtained.

In digestate/control RR, one observation of RR = 14.1 from Cristina et al. (2020) [51] and another of RR = 8.5 from Nabel et al. (2018) [71] were highlighted by the influence analysis due to their high RR values and influence on the pooled result. Consequently, they were removed from dataset 1 to produce a more conservative result. It is noticeable that both observations correspond to experiments on sandy substrate, with low hydric retention and poor nutrient content. In these conditions, digestate treatments exhibited a strong positive effect on crop yield with respect to controls, with values even higher than conventional fertilization treatments. These may suggest a high influence of organic matter input in poor substrates. However, it is likely that the extreme RR values were mainly driven by the abnormally low yields obtained in controls.

Publication bias analyses were either not conclusive or non-significant, indicating no concrete evidence of publication bias. The results of digestate yields compared to control and synthetic fertilizers are considered robust, as significance remained unchanged after modeling conservative, unbiased estimators. Compost tea results also remained unchanged, although they are not reliable due to the low number of observations and consequent limited power of the analyses.

Concerning heterogeneity, the high $I^2$ values indicate that the observations differ in their true effects, which is compatible with the approach of the random-effects model. Pooled effects represent average values obtained from a set of statistical populations [31], whose specific effects are expected to vary due to differences in experiment duration, environment, crop species, measurement techniques, fertilization level, and substrate, among other factors [26,71].

Subgroup analyses of digestate observations were conducted to determine if the results varied depending on the crop species or the environment of the experiments. Yields were statistically similar between digestate and synthetic fertilizer treatments for all the assessed crops except barley. In contrast, yields obtained with digestate were higher than the control treatments' for all crops except forage, suggesting that some crops, such as wheat and lettuce, responded better to fertilization than others. However, the reliability of individual crop results is low as the number of observations was 5 or fewer for all crops except wheat and corn. According to Harrer 2019 [29], it is not recommended to compare two or more subgroups when the entire number of studies in the meta-analysis is smaller than k = 10.

On the other hand, results grouped per environment of the experiment are robust, with k ≥ 18 observations in all cases. As in the general analyses, digestate yields were higher than the control and similar to synthetic fertilizer treatments for both greenhouse and field experiments. The results did not differ significantly between environments, although greenhouse experiments presented higher pooled values.

A meta-analysis of organic fertilizers found that the nitrogen input level of each treatment conditioned yield RR [27]. In the present analysis, N input in the included papers varied in a range between 100 and 480 kg N ha$^{-1}$, with an average input of 244 kg N ha$^{-1}$,

considering only field studies that explicitly reported this value. This factor was not considered as an explanatory variable in the meta-analysis, mainly due to the limited number of observations available.

Compost tea and digestate are produced by the biological transformation of a wide range of organic materials. Thus, fertilizer quality is expected to vary depending on the raw materials used, stabilization level, production method, storage conditions, etc., which may affect crop yield [20,48]. As observed by Luo et al. (2018) [1], experiment duration. may influence on the results due to two facts; first, if an experiment focuses on incomplete crop cycles (e.g., initial stages), treatment differences may not be representative of the total cycle. Second, organic fertilizers contain a high proportion of nutrients in organic forms that require mineralization for plant intake; in this case, short experiments would fail to reflect the effect of long-term nutrient availability [78].

Regarding previous meta-analyses that calculated the effect of organic fertilizers on yield, most or all of the published papers do not focus individually on the effect of compost tea or digestate fertilization. They analyze the effects of a wide range of organic fertilizers grouped in a few classes, including combined application of bioproducts with synthetic fertilizers in most cases. As an example, Ding et al. (2018) [26] found that the combined application of organic and conventional fertilizers increased the yield of rice by 7.8%, with respect to exclusively conventional fertilization. They also suggested that the substitution of mineral N by organic N should not exceed 20% of total N input to avoid yield loss. Other meta-analyses conducted by Liu et al. (2021) [27] found that crop yields increased by 5.5–5.6% with substitution levels lower than 70% of total N.

Luo et al. (2018) [1] and Wang et al. (2020) [2] reported crop yield increase of 27% and 24–28%, respectively, for the combined application of mineral fertilizers and bioproducts, with respect to pure mineral fertilizers. Nonetheless, these studies inform a significant reduction in yields when replacing mineral N by equal amounts of N in organic fertilizers, attributed to a lower bioavailability of this element in the organic matter, which requires microbial degradation. In this sense, Seufert (2019) [78] states that yields of organic cultures are approximately 19–25% lower than those of conventional crops.

The present study provides different results than previous meta-analyses for the comparison between organic fertilizers and synthetic fertilization. Digestate treatments resulted in a pooled yield similar to that obtained with synthetic fertilizers, as the 2% difference between both treatments was not significant. Compost tea produced yields 10% lower than conventional fertilizers, with non-significant difference, based on a very small number of observations. The discrepancies with respect to other meta-analyses may be explained because previous studies considered the effect of different organic products jointly. The analysis of several organic products in one single group may result in loss of information, masking the effect of bioproducts with high fertilization quality in a general average result.

Digestate and compost tea are products of anaerobic and aerobic biodigestion, respectively. Both fertilizers contain a high proportion of nutrients in mineral form, readily available for plant intake in the short term, explaining the similarity with inorganic fertilizers [53,79,80]. These products also provide phytohormones, beneficial microorganisms and slow liberation nutrients contained in the organic matter, and produce a positive effect on soil physical properties, unlike synthetic fertilizers [10,11,61,62,64].

The application of these fertilizers contributes to increasing crop yields beyond the short term, by adding nutrients, beneficial microorganisms, phytohormones and organic matter to soils and substrates [20,61,81]. Besides the plant growth stimulation effect, several authors report pathogen and disease control properties, mainly explained by the presence of biocontrol agents, antibiosis and other mechanisms [14,22,82,83].

Additionally, these products can be combined with both organic and synthetic fertilizers, complementing their different fertilizing properties (e.g., short- or long-term fertilization, organic matter, microbiota, etc.). Combined use with synthetic fertilizers can help

to increase or maintain yields while reducing the input of synthetic products, minimizing environmental negative impacts, as happens with other organic fertilizers [2,27].

Furthermore, production of these organic fertilizers is practicable at different scales. Aerobic and anaerobic digestions may be suitable for both small farms and agro-industrial facilities. Additional benefits include exploiting and treating organic residues while generating high quality products [84,85]. Digestate is a byproduct of biogas, a valuable energetic resource; concurrently, its use as fertilizer improves nutrient cycling and C fixation in agroecosystems [86,87].

The present meta-analysis indicates that digestate is a bioproduct with high potential as fertilizer. Compost tea showed promising results, though further analyses based on a larger number of observations are required to reach more reliable conclusions. Differences in pooled RR between compost tea and digestate were not significant in any of the comparisons (organic fertilizer/control and organic fertilizer/synthetic fertilizer). However, this apparent similarity between the pooled effects of both organic fertilizers is not considered conclusive, due to the poor precision of the compost tea results.

More studies about bioproducts are needed to evaluate crop yield in different scenarios, under diverse environmental conditions and management techniques, in order to establish proportions of raw materials, elaboration processes, and application techniques and doses, ensuring a secure production. In this sense, it is essential that the results of future experimental research be published with all the information necessary for their inclusion in future meta-analyses, as previously detailed. Finally, meta-analysis proved to be a powerful tool to summarize scientific information, reaching strong and comprehensive conclusions.

## 6. Conclusions

According to this study, fertilization with digestate significantly increased yields with respect to control, reaching values similar to conventional fertilizers. This result is superior to those reported by other meta-analyses for organic fertilizers, as previous works grouped the effects of diverse organic products in a few categories, probably pooling together the effects of high-performance organic fertilizers with those of poorer value.

The results for digestate are considered robust as their significance did not vary even after modeling conservative, "unbiased" estimators. However, the high heterogeneity observed suggests the existence of factors that may account for part of the variability. Although the influence of crop species and environment of the experiments was assessed by the application of subgroup analysis, further analyses based on a larger number of studies are required in order to obtain a robust estimation of the effect of crop species and other variables on yields.

In the case of compost tea, the analysis indicates that yields can be significantly higher than the control and lower than that of synthetic fertilizers, with non-significant difference. Nevertheless, these results are considered merely informative and further analyses are required as the compost tea dataset comprised a very low number of observations. "Unbiased" estimators for compost tea showed no change in the results, although they are not reliable due to the low number of observations.

To conclude, meta-analysis proved to be a powerful tool that makes it possible to summarize scientific information, reaching strong and comprehensive conclusions about organic fertilization. However, it is required that experimental studies inform dispersion values for each independent treatment, thus reducing the loss of information in posterior meta-analyses and increasing reliability of the results.

**Author Contributions:** Conceptualization, F.C. and I.F.-P.; methodology, F.C. and M.A.; software, F.C. and M.A.; validation, F.C., E.M.U. and I.F.-P.; formal analysis, F.C.; investigation, F.C., E.M.U. and I.F.-P.; resources, I.F.-P., E.M.U. and M.C.; data curation, F.C.; writing—original draft preparation, F.C. and I.F.-P.; writing—review and editing, E.M.U. and M.C.; visualization, E.M.U. and M.C.; supervision, I.F.-P., E.M.U. and M.C.; project administration, I.F.-P.; funding acquisition, I.F.-P. All authors have read and agreed to the published version of the manuscript.

**Funding:** This work was supported by the Fondo para la Investigación Científica y Tecnológica, Ministerio de Ciencia, Tecnología e Innovación, Argentina, under Grant PICT 2019-2193 to Iván Funes-Pinter.

**Institutional Review Board Statement:** Not applicable.

**Informed Consent Statement:** Not applicable.

**Data Availability Statement:** Not applicable.

**Conflicts of Interest:** The authors declare no conflict of interest.

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
