# Peer review of "Meta-Analysis of Yields of Crops Fertilized with Compost Tea and Anaerobic Digestate"

_sustainability, doi:10.3390/su15021357_

Round 1
Reviewer 1 Report
The authors were demonstrated the information in a systematic way with good number of citation. This representation of the topic is very basic, interesting and can attract a lot of attention. It is also necessary to critically evaluate and do not make hasty conclusions, which may lead to misinterpretations. Several points are important to be addressed before going to possible publication in this high-quality journal. In addition, the authors need to address all points in the revision stage for broad range readers understanding.
1. Abstract: Please keep in mind that this section is completely different from the Introduction section. Authors are suggested to mention the novelty of the study clearly. The authors need to consider these points in the revision stage.
2. The Introduction section needs to be extended with describing the cause, fatality, novelty of the work, advantages and disadvantages of the present study.
3. Therefore, the abstract and introduction needs to be improved. The systematic bibliographic search section, keywords representation could be better. Please modify.
4. Equation number and citations are missing. Kindly update.
5. Barring a few sentences in the text, the English language is fair. However, the text is not free from grammar errors. Ensure that your English manuscript is guaranteed free of language issues. In addition, the manuscript should be thoroughly checked for English corrections as there are some colloquial terms being used.
6. Citation format needs to be checked carefully according to the journal guideline, For example: Line : 295
7. Fig 3, Y axis title is missing. Kindly update all the fig accordingly.
8. Line: 450. Kindly look into the word “Latter”

Author Response
Thank you for your time and valuable suggestions. We consider all of your comments:
- Abstract: Please keep in mind that this section is completely different from the Introduction section. Authors are suggested to mention the novelty of the study clearly. The authors need to consider these points in the revision stage.
R: Suggestions were incorporated
- The Introduction section needs to be extended with describing the cause, fatality, novelty of the work, advantages and disadvantages of the present study.
R: Done according suggestions
- Therefore, the abstract and introduction needs to be improved. The systematic bibliographic search section, keywords representation could be better. Please modify.
R: Done
- Equation number and citations are missing. Kindly update.
R: Equations number and citation were included
- Barring a few sentences in the text, the English language is fair. However, the text is not free from grammar errors. Ensure that your English manuscript is guaranteed free of language issues. In addition, the manuscript should be thoroughly checked for English corrections as there are some colloquial terms being used.
R: English was revised
- Citation format needs to be checked carefully according to the journal guideline, For example: Line : 295
R: Tha manuscript was formatted according guideline
- Fig 3, Y axis title is missing. Kindly update all the fig accordingly.
R: Done
- Line: 450. Kindly look into the word “Latter”
R: Done
Please see the attachment with all reviewers comments incorporated, in general highlighted in yellow.
Reviewer 2 Report
This research topic was meaningful, and workload is enough. Some suggestions were given as below:
(1) Line 24: “Yields were statistically similar to those obtained with conventional treatments (only 2% lower in average)”, conventional treatments should be conventional treatment with chemical fertilization.
(2) Line 27: The authors should state if the high heterogeneity level affects the result reliability of this meta-analysis.
(3) Line 29: The more specific results of “anaerobic digestate had good potential as fertilizer in agriculture” should be given, e.g., the application rate range of digestate should be included.
(4) Keywords: the response ratio should be substituted with meta-analysis, and compost tea should be included.
(5) Line 85-91: The authors should strengthen that if there are any review or meta-analysis about the fertilization effects of compost tea and biogas on crop yield? They should be included.
(6) Line 102-121: It’s unnecessary to list all the keywords here, please reorganize them by referring to doi: 10.1007/s11356-022-19909-1, 10.1016/j.scitotenv.2020.144058.
(7) Line 242-323: Too many common knowledge were given here, please shorten this part, the authors can referring to above mentioned and other meta-analysis.
(8) Line 336-344: The cases in the dataset was small, the authors no needs to strengthen this result here. However, the authors should combine the meta-analysis results with those publication bias in following discussion.
(9) Line 414: “suggesting the lack of publication bias”, how to draw this conclusion?
(10) Line 455-461: Critical discussion of meta-analysis should be included; current paragraph was not important for this meta-analysis topic.
(11) Subgroup analysis should be included, by referring the above two recommended papers.
Author Response
Thank you for your time and valuable suggestions. We consider all of your comments:
(1) Line 24: “Yields were statistically similar to those obtained with conventional treatments (only 2% lower in average)”, conventional treatments should be conventional treatment with chemical fertilization.
R: Done
(2) Line 27: The authors should state if the high heterogeneity level affects the result reliability of this meta-analysis.
R: Done
(3) Line 29: The more specific results of “anaerobic digestate had good potential as fertilizer in agriculture” should be given, e.g., the application rate range of digestate should be included.
R: Suggestion was considered
(4) Keywords: the response ratio should be substituted with meta-analysis, and compost tea should be included.
R: The title already includes those terms, we understand that they should not be repeated in both
(5) Line 85-91: The authors should strengthen that if there are any review or meta-analysis about the fertilization effects of compost tea and biogas on crop yield? They should be included.
R: Some paragraphs about it were included in the “Introduction” section
(6) Line 102-121: It’s unnecessary to list all the keywords here, please reorganize them by referring to doi: 10.1007/s11356-022-19909-1, 10.1016/j.scitotenv.2020.144058.
R: Done
(7) Line 242-323: Too many common knowledge were given here, please shorten this part, the authors can referring to above mentioned and other meta-analysis.
R: Done
(8) Line 336-344: The cases in the dataset was small, the authors no needs to strengthen this result here. However, the authors should combine the meta-analysis results with those publication bias in following discussion.
R: The manuscript was improve according the suggestion
(9) Line 414: “suggesting the lack of publication bias”, how to draw this conclusion?
R: Some text clarifying this was included in the manuscript
(10) Line 455-461: Critical discussion of meta-analysis should be included; current paragraph was not important for this meta-analysis topic.
R: According suggestion, more discussion was included
Please see the attachment with all reviewers comments incorporated, in general highlighted in yellow.
Reviewer 3 Report
The paper" Meta-analysis of yields of crops fertilized with compost tea and anaerobic digestate" presents a meta-analysis of existing scientific evidence. The paper is good but contains significant amount of flaws therefore the paper is unacceptable in current form
1. Please follow the journal guidelines in formating article as there are some errors in current format like you have bolded the references in text. please make corrections
2.1.Please revise the introduction by adding 1-2 paragraphs on adverse impacts of conventional fertilizers on soil globally and advantages of using organic fertilizer.
3. What is the novelty of this study? Describe it at the end of introduction.
4. Don’t you think that the study is more hypothetical and it may have no significance if it is done in real way? Why you did not carry out the physical studies by putting field trials?
5. Please compare the results of your study with other similar studies wherever needed
6.Conclusion should be revised and shortened by including major findings only.
7. Kindly include some references from the journal of sustainablity mdpi
Author Response
Thank you for your time and valuable suggestions. We consider all of your comments:
- Please follow the journal guidelines in formating article as there are some errors in current format like you have bolded the references in text. please make corrections
R: The manuscript has been formatted according guideline
2. Please revise the introduction by adding 1-2 paragraphs on adverse impacts of conventional fertilizers on soil globally and advantages of using organic fertilizer.
R: Some sentences were modified and added according to suggestion
- What is the novelty of this study? Describe it at the end of introduction.
R: Done
- Don’t you think that the study is more hypothetical and it may have no significance if it is done in real way? Why you did not carry out the physical studies by putting field trials?
R: The purpose of a meta-analysis is different than that of individual trials. A meta-analysis intends to summarize existing scientific evidence to obtain general conclusions from multiple existing observations (published experiments). While one field trial would determine the effect of organic fertilizer for a certain crop under specific conditions, the contribution of this meta-analysis is to provide an average value from a wide range of conditions.
- Please compare the results of your study with other similar studies wherever needed
R: In “Discussion” section we approach about it, and we added some new sentences
6.Conclusion should be revised and shortened by including major findings only.
R: Done
- Kindly include some references from the journal of sustainablity mdpi
R: Done
Please see the attached document with all reviewers comments incorporated, in general highlighted in yellow.
Round 2
Reviewer 2 Report
It’s appreciated that the authors have make great improvement to this manuscript. However, current manuscript still remains many problems. As a meta-analysis, the subgroup analysis is needed, which was lacked in current review, even the authors didn’t calculate the general trend in current manuscript. In addition, the normal concepts for meta-analysis account in Line 224 - Line 320 (c.a. 20% of total manuscript), but the key discussion (in Line 435-559) only account < 20%. So, it’s obvious that current form only can present simple data description, but the impressive conclusion for readership.
Author Response
Thak you for your suggestions. They have been incorporated, please find them highlighted in the main document

Reviewer 3 Report
The paper is revised well and can be published in current form
Author Response
Thak you for your time and comments.
Round 3
Reviewer 2 Report
There are some minor problems needed to be addressed before publication:
(1) "480 kgN ha-1 " should be changed to "480 kg N ha-1 ".
(2) The introduction and conclusion should include or breifly introduce the subgroup analysis results.
Author Response
Thank you for your detailed evaluation. We incorporated all suggestions.